# Diagnosing Transformers: Illuminating Feature Spaces for Clinical Decision-Making

**Aliyah R. Hsu**
Department of EECS
UC Berkeley
aliyahhsu@berkeley.edu

**Yeshwanth Cherapanamjeri**
CSAIL, MIT

**Briton Park**
Department of Statistics
UC Berkeley

**Tristan Naumann**
Microsoft Research

**Anobel Y. Odisho**
Department of Urology, Epidemiology and Biostatistics
UC San Francisco

**Bin Yu**
Department of Statistics, EECS
Center for Computational Biology
UC Berkeley

## Abstract

Pre-trained transformers are often fine-tuned to aid clinical decision-making using limited clinical notes. Model interpretability is crucial, especially in high-stakes domains like medicine, to establish trust and ensure safety, which requires human engagement. We introduce SUFO, a systematic framework that enhances interpretability of fine-tuned transformer feature spaces. SUFO utilizes a range of analytic and visualization techniques, including Supervised probing, Unsupervised similarity analysis, Feature dynamics, and Outlier analysis to address key questions about model trust and interpretability (e.g. model suitability for a task, feature space evolution during fine-tuning, and interpretation of fine-tuned features and failure modes). We conduct a case study investigating the impact of pre-training data where we focus on real-world pathology classification tasks, and validate our findings on MedNLI. We evaluate five 110M-sized pre-trained transformer models, categorized into general-domain (BERT, TNLR), mixed-domain (BioBERT, Clinical BioBERT), and domain-specific (PubMedBERT) groups. Our SUFO analyses reveal that: (1) while PubMedBERT, the domain-specific model, contains valuable information for fine-tuning, it can overfit to minority classes when class imbalances exist. In contrast, mixed-domain models exhibit greater resistance to overfitting, suggesting potential improvements in domain-specific model robustness; (2) in-domain pre-training accelerates feature disambiguation [1] during fine-tuning; and (3) feature spaces undergo significant sparsification during this process, enabling clinicians to identify common outlier modes among fine-tuned models as demonstrated in this paper. These findings showcase the utility of SUFO in enhancing trust and safety when using transformers in medicine, and we believe SUFO can aid practitioners in evaluating fine-tuned language models (LMs) for other applications in medicine and in more critical domains. [2]

## 1 Introduction

Pre-trained transformer models achieve state-of-the-art performance on a range of natural language processing (NLP) tasks (Devlin et al., 2019; Lewis et al., 2020). As a consequence, we have witnessed their increasing adoption in the medical domain (Yalunin et al., 2022; Zhang et al., 2021).

---

[1]We refer to the clustering of the feature space according to the labels of the input datapoint.

[2]Our source code is available at https://github.com/adelaidehsu/path_model_evaluation

While they achieve strong empirical performance, little is understood about how they obtain these results or when they lead to unreliable performance. In these critical applications, understanding into or interpretability (or explainability) of the predictions is indispensable for building trust in these models for medical personnel and patients alike.

In this paper, we propose a systematic framework that provides a practical pipeline for analyzing and interpreting models fine-tuned for particular prediction tasks, focusing on important questions about model trust and interpretability: model suitability for a task, feature space evolution during fine-tuning, and interpretation of fine-tuned features and failure modes. Our framework leverages a suite of analytic and visualization techniques to interpret the feature space of a fine-tuned model.

Distribution shift is a natural consequence of applying general-domain pre-trained models to the medical domain (Gao et al., 2021). Mixed-domain pre-trained models that perform continual pre-training with biomedical data partially address this issue and have demonstrated improved performance on several medical tasks (Gururangan et al., 2020; Grambow et al., 2022; Alsentzer et al., 2019; Lee et al., 2019). Domain-specific models, which are pre-trained *from scratch* with biomedical data, further alleviate this issue by allowing for specialized vocabularies better reflective of the medical domain. Although these models yield improved performance, their vulnerability to spelling mistakes resulting from a highly specialized vocabulary and limited pre-training data is well documented (Bressem et al., 2020; Richter-Pechanski et al., 2021; Lentzen et al., 2022).

We use SUFO to comprehensively investigate the effects of pre-training data distributions for a real-world pathology report dataset, and further support our findings with a public clinical dataset, MedNLI. We evaluate five pre-trained transformer models (Subsection 3.1) of the same size but differing in pre-training corpora (general-domain/mixed-domain/domain-specific) on our five tasks (Subsection 3.2). In this setting, SUFO helps study the following instantiations of its general targets: (1) how much does in-domain pre-training help? (Section 4)? (2) what changes in the feature space during fine-tuning to have led to the differences in model performance (Section 5)? (3) how do we interpret the fine-tuned feature space and analyze their failure modes (Sections 6)?

We call our approach SUFO and explain below where the name SUFO stands for by making the corresponding letters bold. Each component of SUFO was chosen to yield complementary insights into each of these questions. Firstly, **S**upervised probing evaluates model features by directly using them for prediction with minimal fine-tuning and sheds light on the suitability of certain pre-trained model for a target task. We show that although pre-trained features in a domain-specific model may contain the most useful information, a domain-specific model can overfit to minority classes after fine-tuning, when presented with class imbalance, while mixed-domain pre-trained models are more resistant to overfitting.

Secondly, **U**nsupervised similarity analysis and **F**eature Dynamics visualization study the evolution of the learned feature spaces through fine-tuning and qualitatively disambiguate these models both through their speed of convergence and the degree to which they deviate from the pre-trained initialization. We find the benefit of in-domain pre-training is manifested in faster feature disambiguation; however, the key determinants of model performance are the closeness of pre-training and target tasks and a diverse pre-training data source enabling more robust textual modeling.

Finally, through the substantial sparsification of feature spaces induced by fine-tuning, **O**utlier analysis allows for a deeper understanding of the failure modes of these models. We observe that models pre-trained with in-domain data discover a more diverse set of challenging/erroneous reports as determined by a domain expert than a general-domain model.

SUFO may inform the practical use of these models by aiding in the selection of an appropriate pre-trained model, a quantitative and qualitative evaluation of these models through fine-tuning, and finally, an understanding of their failure modes for more reliable deployment.

The rest of the paper is organized as follows. We present related work in Section 2, our experimental setup in Section 3. To formalize SUFO, we describe the use of the Supervised probing in Section 4, while the Unsupervised similarity and Feature Dynamics analysis are performed in Section 5. The Outlier analysis with expert validation is in Section 6, and Section 7 concludes the paper.

## 2 RELATED WORK

**LMs performance on clinical tasks**   Prior work has noted the benefits of including biomedical data in the pre-training corpora (Yalunin et al., 2022; Grambow et al., 2022; Alsentzer et al., 2019; Lee et al., 2019), and the nuances of when and how to include such data (Gururangan et al., 2020). Yet, a comprehensive analysis of the impact of these choices on transformer features remains elusive, and our work aims to provide this understanding to offer improved prescriptive recommendations for practitioners.

In concurrent work, Kefeli & Tatonetti (2023) released a model fine-tuned with ClinicalBERT on pathology reports for primary Gleason score extractions.[3] Tai et al. (2020) adapts BERT to the medical domain by adding a domain-specific embedding layer and extending the vocabulary. Domain-specific models (Gu et al., 2021; Rasmy et al., 2020; Li et al., 2019; Aum & Choe, 2021) are proposed to further mitigate the problem of distribution shifts (Gao et al., 2021) with pre-training using biomedical data only. These models have shown improved performances on biomedical benchmarks (Tinn et al., 2023), and many clinical tasks spanning from medical abstraction (Preston et al., 2022), drug-target interaction identification (Aldahdooh et al., 2022), to clinical classifications (Tejani et al., 2022; Mantas et al., 2021); however, their vulnerability regarding grammatical mistakes is also discussed (Bressem et al., 2020; Richter-Pechanski et al., 2021; Lentzen et al., 2022).

**Feature analysis in LMs**   Most prior works have focused on token feature analysis in unsupervised LM encoders. Supervised probing models, or diagnostic classifiers, are widely used in such works to test features for linguistic phenomena (Tenney et al., 2019; Liu et al., 2019a; Peters et al., 2018) and syntactic structure (Hewitt & Manning, 2019). With increased flexibility, unsupervised techniques are also proposed to investigate features in the same encoders. SV-CCA (Raghu et al., 2017), a form of canonical correlation analysis, is used in a cross-temporal feature analysis for learning dynamics (Saphra & Lopez, 2019), while PW-CCA (Morcos et al., 2018), an improved version of SV-CCA, is used to analyze transformer features under different pre-training objectives (Voita et al., 2019). RSA (Kriegeskorte et al., 2008) is increasingly used, such as in investigating the sensitivity of features to context (Abnar et al., 2019), and the correspondence of natural language features to syntax (Chrupała & Alishahi, 2019). In addition to the works performed on unsupervised LMs, our work builds on a line of recent works focusing on the fine-tuning effect on BERT for natural language understanding (NLU) tasks. Peters et al. (2019) discussed the choice of adaptation methods based on the performance of task-specific probing models at various layers. van Aken et al. (2019) interpreted question-answering models through cluster analysis. Structural probing, RSA and layer ablations are also used in investigating the fine-tuning process of BERT (Merchant et al., 2020), and correlating features of a fine-tuned BERT to fMRI voxel features (Gauthier & Levy, 2019).

Our work improves upon feature analysis since it integrates feature analyses to enable enhanced interpretability of the fine-tuned feature spaces of transformer, and provides insights into the impact of pre-training data on fine-tuned transformers features. This integrated feature analysis pipeline SUFO allows a clearer window through which the inner workings of fine-tuned LMs become more accessible to domain experts such as clinicians. Such domain expert engagements are indispensable for building trust in LMs and ensuring their safety in medicine.

## 3 EXPERIMENTAL SETUP

### 3.1 PRE-TRAINED MODELS

We evaluate five 110M-sized [4] encoder-based [5] transformer (Vaswani et al., 2017) models commonly used in clinical classifications. Here we describe the models, with an emphasis on their differences in pre-training objectives and categories of pre-training corpora.

---

[3]This concurrent model isn't included in our evaluation due to time constraints before submission deadlines.

[4]Models exhibit only small changes in vocabulary sizes (28996-30522) and the 110M parameter counts include the sizes of the word embeddings.

[5]We focus our discussion on strictly encoder-based transformers by not considering transformers in other architectures (i.e. decoder-only, encoder-decoder) to avoid introducing extra confounding factors.

**General-domain: BERT and TNLR**   The popular BERT (Devlin et al., 2019) architecture is based on bidirectional transformer encoder (Vaswani et al., 2017). BERT is pre-trained on masked language-modeling (MLM) and next sentence prediction tasks, with a general-domain corpus (3.3B words) from BooksCorpus (Zhu et al., 2015) and English Wikipedia. We use BERT$_{\text{BASE}}$ with 12 layers and 12 attention heads, and the uncased WordPiece (Wu et al., 2016) tokenization since prior work (Gu et al., 2021) has established that case does not have a significant impact on biomedical downstream tasks. The Turing Natural Language Representation (TNLR) model (Bao et al., 2020) we use has the same architecture and vocabulary as BERT. They do differ, however, in their pre-training objectives, self-attention mechanism, and data as TNLR is trained using constrained self-attention with a pseudo-masked language modeling (PMLM) (Bao et al., 2020) task on a more diverse general-domain corpus (160GB) that additionally includes OpenWebText[6], CC-News (Liu et al., 2019b), and Stories (Trinh & Le, 2018).

**Mixed-domain: BioBERT and Clinical BioBERT**   BioBERT (Lee et al., 2019) and Clinical BioBERT (Alsentzer et al., 2019) are categorized as mixed-domain pre-trained models because they are pre-trained with biomedical data on top of a general-domain corpus. The version we use is obtained via continual pre-training from BERT by training on PubMed abstracts (4.5B) for additional steps. Clinical BioBERT is the result of continual pre-training from BioBERT by training additionally on MIMIC-III clinical notes (0.5B) to be more tailored for clinical tasks. The two models share the same vocabulary and architecture as BERT.

**Domain-specific: PubMedBERT**   PubMedBERT (Gu et al., 2021) was proposed to mitigate the shortcomings in BERT's vocabulary as it cannot represent biomedical terms in full, which was found to possibly hinder the performance of general-domain and mixed-domain models on downstream biomedical tasks (Tinn et al., 2023; Tai et al., 2020; Gao et al., 2021). Hence, this model is trained from scratch using PubMed abstracts (3.1B) only, resulting in a more specialized vocabulary for biomedical tasks. We use the uncased version of PubMedBERT with the same architecture as BERT.

**Remark on differences in pre-training objectives and data**   The pre-training objectives and data sizes are similar for all the BERT-based models and we do not expect these differences to impact our findings. While TNLR has a different objective and self-attention mechanism which could confound our analysis, we find that the quantitative and qualitative behavior observed in its analysis in relation to the mixed and domain-specific models are similar to BERT, the other general-domain model. Thus, we believe that our conclusions are applicable to TNLR despite these differences.

## 3.2   FINE-TUNING DATA

**Prostate Cancer Pathology Reports**   We collected a corpus of 2907 structured pathology reports with data elements extracted from a set of free-text reports following a previously proposed preprocessing pipeline (Odisho et al., 2020). The corpus includes pathology reports for patients that had undergone radical prostatectomy for prostate cancer at the University of California, San Francisco (UCSF) from 2001 to 2018. This study was conducted under an institutional review board (IRB) approval. The reports contain an average of 471 tokens. For each document, we focus on the following 4 pathologic data elements: primary Gleason grade (Path-PG), secondary Gleason grade (Path-SG), margin status for tumor (Path-MS), and seminal vesicle invasion (Path-SV), and formed 4 classification tasks correspondingly. (Detailed description in Appendix A.1) For Path-PG and Path-SG, there are 5 labels available: [null, 2, 3, 4, 5], with null denoting an undecided Gleason score, often due to previous treatment effects. We exclude reports with null and 2 Gleason scores under a doctor's suggestion as the two labels account for only 1.3% and 0.07% of the corpus, and are rarely graded in practice. [7]  After the removal, the distribution of labels 3, 4, and 5 in Path-PG is 67%, 30%, and 3% respectively, while in Path-SG it is 39%, 53%, and 8%. Both Path-MS and Path-SV are binary classification tasks, with only two labels: [positive, negative]. The distribution of positive and negative in Path-MS is 26% and 74%, while in Path-SV is 13% and 87%.

---

[6]skylion007.github.io/OpenWebTextCorpus

[7]Previously we tried to include Gleason scores null and 2 in the fine-tuning, but found none of the models could classify any of the two classes well due to their extremely small sample sizes. It didn't seem reasonable to discuss the models' performance on these two classes given that they couldn't even learn well.

Our pathology reports dataset is not publicly available due to the protected patient information in the dataset; however, we provide a few anonymized report samples in Appendix A.2 as illustration.

**MedNLI**  To support the generalizability of our conclusions, we additionally report the fine-tuning results of the models on a publicly available clinical dataset, MedNLI (Romanov & Shivade). The objective of MedNLI is to determine if a given clinical hypothesis can be inferred from a given premise, and the dataset is labelled with three classes [contradiction, entailment, neutral]. We (non-uniformly) sample subsets of 6990 samples from MedNLI which reflect the different class distributions observed in the pathology report extraction tasks.

See Appendix A.3 for a full description of fine-tuning hyperparameters. Note that random weighted sampling was implemented for all tasks during fine-tuning to tackle the data imbalance.

## 4  HOW MUCH DOES IN-DOMAIN PRE-TRAINING HELP?

In this section, we discuss realistic scenarios when in-domain pretraining [8] benefits, and more importantly, hinders, downstream task performances by analyzing performance of the pre-trained models under two most common forms of adaptation: fine-tuning and supervised probing.

### 4.1  MODEL PERFORMANCE: FINE-TUNING

We show the fine-tuned model performance on pathology reports in Table 1. The models have generally comparable performance on Path-SG, Path-MS, and Path-SV; however, they are distinguished by their performance on Path-PG, where serious data imbalance exists. In Path-PG, BioBERT and Clinical BioBERT still obtain relatively high accuracies, $> 93\%$, while classifying both majority *and* minority classes well (see Appendix A.4 for per-class accuracy). The general-domain models, BERT and TNLR, having accuracies 86% and 76% on Path-PG, show inferior performance to the mixed-domain models. Yet surprisingly, PubMedBERT, as a domain-specific model, also does poorly on Path-PG performing close to the general-domain models. Specifically, we find that while PubMedBERT does well on the majority classes, it struggles with the minority one.

To investigate whether this finding extends outside of our pathology report dataset, we evaluated the fine-tuning performance of PubMedBERT and Clinical BioBERT on MedNLI, where we simulated three scenarios of different class distributions: Balanced, Imbalanced (simulating class distribution in Path-SG), and Highly Imbalanced (simulating class distribution in Path-PG), and report the results in Table A3. In the Balanced set, PubMedBERT can outperform Clinical BioBERT. However, Clinical BioBERT outperforms PubMedBERT in the Highly Imbalanced set due to PubMedBERT's inability to classify one of the minority groups well, while in the Imbalanced set, both yield comparable performance, corroborating our finding on the pathology reports. Hence, for the feature analyses in the following sections, we will focus on the pathology report dataset.

### 4.2  MODEL PERFORMANCE: SUPERVISED PROBING

Supervised probing, where we freeze the pre-trained weights, and only train the last linear layer, is a measure of how much useful information for a downstream task is contained in the pre-trained features (van Aken et al., 2019; Peters et al., 2019; Merchant et al., 2020; Hewitt & Manning, 2019). We report the supervised probing performance on pathology reports in Table 2. For comparison, we provide baseline results on a randomly initialized BERT (Random-BERT). This normalization is necessary as even random features often perform well in probing methods. (Zhang & Bowman, 2018; Hewitt & Liang, 2019). Among all, PubMedBERT achieves the highest average score while the mixed-domain models and BERT, come second with average scores close to PubMedBERT, and TNLR obtains the lowest average score failing to even beat the baseline.

---

[8] In-domain pre-training includes both mixed and domain-specific pre-training, as long as biomedical data is included in the pre-training data.

Table 1: F1 test set performance over 3 runs. BioBERT and Clinical BioBERT perform the best on average, while PubMedBERT struggles when serious data imbalance present.

| Models | Path-PG | Path-SG | Path-MS | Path-SV | Average |
|---|---|---|---|---|---|
| BERT | 0.858 (0.16) | 0.975 (0.02) | 0.957 (0.01) | 0.908 (0.03) | 0.924 |
| TNLR | 0.763 (0.18) | 0.995 (0.01) | 0.963 (0.01) | 0.932 (0.01) | 0.913 |
| BioBERT | 0.933 (0.04) | 0.991 (0.01) | 0.959 (0.01) | 0.915 (0.02) | 0.950 |
| Clinical BioBERT | 0.959 (0.03) | 0.992 (0.01) | 0.964 (0.01) | 0.920 (0.01) | 0.959 |
| PubMedBERT | 0.770 (0.12) | 0.984 (0.01) | 0.970 (0.01) | 0.928 (0.01) | 0.913 |

Table 2: F1 test set performance under supervised probing over 3 runs. PubMedBERT performs the best, showing its pre-trained feature contains the most useful information for pathology reports.

| Models | Path-PG | Path-SG | Path-MS | Path-SV | Average |
|---|---|---|---|---|---|
| BERT | 0.371 (0.04) | 0.345 (0.05) | 0.678 (0.03) | 0.578 (0.02) | 0.493 |
| TNLR | 0.271 (0.01) | 0.267 (0.06) | 0.494 (0.03) | 0.481 (0.01) | 0.378 |
| BioBERT | 0.340 (0.04) | 0.327 (0.04) | 0.666 (0.03) | 0.574 (0.02) | 0.477 |
| Clinical BioBERT | 0.341 (0.03) | 0.336 (0.04) | 0.689 (0.02) | 0.581 (0.01) | 0.487 |
| PubMedBERT | 0.387 (0.01) | 0.327 (0.03) | 0.687 (0.02) | 0.575 (0.01) | 0.494 |
| Random-BERT | 0.339 (0.06) | 0.260 (0.09) | 0.529 (0.05) | 0.555 (0.05) | 0.421 |

### 4.3 DISCUSSION ON EFFECT OF IN-DOMAIN PRE-TRAINING DATA

The results in Subsections 4.1 and 4.2 demonstrate some subtle effects of in-domain pre-training data when it brings performance gain. That is, even under different degrees of class imbalance, *if* pre-training data is diverse enough to ensure robustness, the gain persists. PubMedBERT is shown to contain much useful information for our tasks in its pre-trained features, possibly due to its domain-specific pre-training; however, it suffers from instability in predicting the minority class after fine-tuning. Mixed-domain models, such as BioBERT and Clinical BioBERT, not only show good performance on supervised probing, but also perform well after fine-tuning. The benefits of their mixed-domain pre-training are two-fold: first, pre-training on biomedical datasets allows for better domain-specific features more amenable to performance improvements through fine-tuning, and second, the incorporation of general-domain corpus makes them more resistant to overfitting.

## 5 WHAT HAPPENS DURING FINE-TUNING?

In Sec 4, we observe how fine-tuning distorts the features in PubMedBERT, making it no longer the most suitable for the pathology classification tasks after fine-tuning. This indicates a significant change in feature space through the fine-tuning process, and we investigate this change across layer and time in this section. We first leverage an unsupervised similarity analysis, to measure similarity of content of neural representations (Laakso & Cottrell, 2000) across layers. Next, we explore feature dynamics, along both time and layer axes, through cluster analysis, to examine the structure and evolution of feature disambiguation during fine-tuning. Our work is, to our knowledge, the first to conduct such extensive cluster analysis on text features, as previous studies often focus on either cross-layer or cross-temporal analysis. (van Aken et al., 2019; Voita et al., 2019)

### 5.1 UNSUPERVISED REPRESENTATIONAL SIMILARITY ANALYSIS (RSA): CHANGES IN THE FEATURE SPACE AFTER FINE-TUNING

RSA is an unsupervised technique for measuring the similarity of two different feature spaces given a set of control stimuli. It was first developed in neuroscience (Kriegeskorte et al., 2008), and has been increasingly used to analyze similarity between neural network activations (Merchant et al., 2020; Abnar et al., 2019; Chrupała & Alishahi, 2019). To conduct RSA, a common set of $n$ samples is used to create two sets of features from two models separately. For each feature set, a pairwise similarity matrix in $\mathbb{R}^{n \times n}$ is calculated with a defined distance measure. The final similarity score between the two feature spaces is computed as the Pearson correlation between the flattened upper

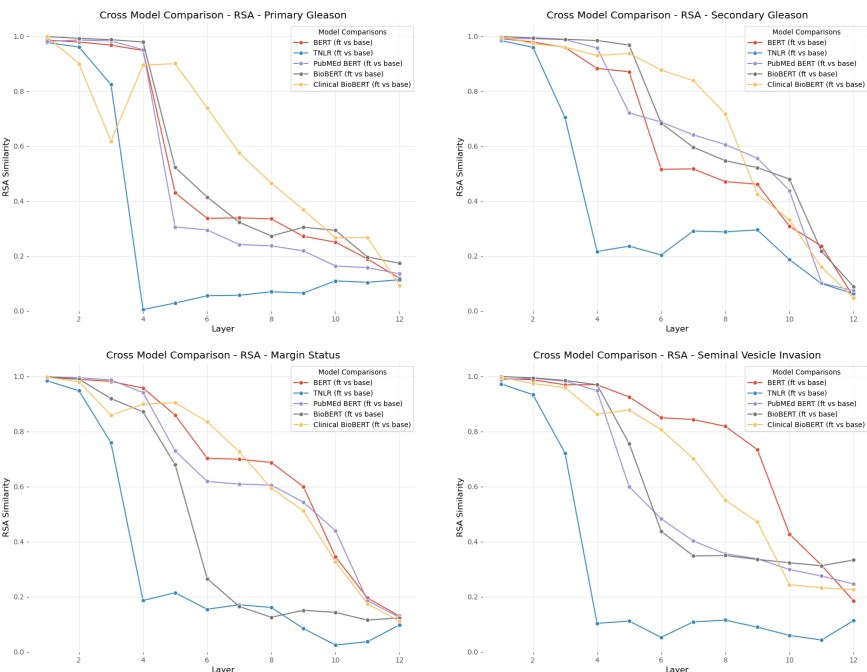

Figure 1: Layer-wise RSA comparing the pre-trained and fine-tuned versions of the models across four pathology classification tasks.

triangular sections of the two pairwise similarity matrices. In our work, we sample random reports ($n = 1000$) from our dataset for each of the four tasks as the control stimuli. We extract activations of corresponding encoder layers at the classification token from the two versions, e.g. pre-trained vs. fine-tuned, of each model as the feature sets to compare, in an effort to examine the layer-wise change brought by the fine-tuning process. We use Euclidean distance as the defined distance measure to calculate the pairwise similarity matrix.[9]

**Results**   Figure 1 shows our RSA results comparing the pre-trained and fine-tuned versions of each of the five models. In the figure, lower values imply greater change relative to the pre-trained model. We observe a few common trends across all tasks. First, the changes generally arise in the middle layers of the network, and increase in the layers closer to the loss, with little change observed in the layers closest to the input, possibly due to vanishing gradient. Second, Clinical BioBERT on average has the smallest change across layers, or retains the most pre-trained information, while TNLR undergoes the most drastic reconfiguration, suggesting Clinical BioBERT having the pre-trained data distribution more aligned to our target task which are less distorted during fine-tuning, while that of TNLR is the most distant[10]. On average, BERT, BioBERT and PubMedBERT show moderate reconfiguration in the layers, which especially indicates the versatility of BERT's feature space for its ability to match models pre-trained using in-domain data with relatively little reconfiguration.

## 5.2   FEATURE DYNAMICS: CLUSTER ANALYSIS ACROSS LAYER AND TIME

We use PCA to investigate feature dynamics in the models. We examine the structure and evolution of feature disambiguation during fine-tuning across two axes: layer and time. By examining whether feature disambiguation coincides with layers shown to change the most in Subsection 5.1, we are able to discuss how much of the change actually translates into useful information for the tasks. We extract activations of corresponding encoder layers at the classification token across all 25 checkpoints as feature sets used in this experiment.

---

[9]We experimented with both Euclidean distance and cosine similarity, and in practice not much difference was observed between the results.

[10]See Appendix A.6 for a quantitative definition of the closeness between pre-training data and target data.

**Results**   Due to space limitations, we present test set feature dynamics of the five models in Appendix A.9, where we include the results from Path-PG and Path-MS, as representatives of tasks having different number of labels, as we observe similar results across all the four tasks. When comparing the feature dynamics with RSA results in Subsection 5.1, we observe that the change in layers measured using RSA typically translate into useful information for target tasks, as we see feature disambiguation in layers often coincides with the significant drop in RSA scores for all the five models. The feature dynamics of TNLR is generally the most dissimilar with the rest. For example, in Path-PG, TNLR disambiguates the minority class, 5, first starting in layer 4, and then the majority classes, 3 and 4, starting in layer 7; however, we observe the opposite behavior in the remaining BERT-based models, where they disambiguate the majority class before the minority class. We suspect this difference results from the pre-training objectives and self-attention mechanisms. Examining the feature dynamics through time, we do see models leveraging in-domain pre-training, such as BioBERT, Clinical BioBERT, and PubMedBERT, disambiguate faster than general-domain models. In Path-PG, in-domain models start to disambiguate the classes at around epoch 6, while BERT and TNLR do so around epoch 9. Overall, Clinical BioBERT requires the fewest change in layers and less training epochs to disambiguate the features well. The mixing of classes shown in PubMedBERT's scatterplots on Path-PG, which was not observed in its train set feature dynamics, corroborates the overfitting problem that we see in its fine-tuning performance. From the result, we argue that the effect of in-domain pre-training is manifested in less fine-tuning epochs needed, but the quality of final feature disambiguation really affecting a model's performance is dependent on the closeness of pre-training and target tasks, and the model's resistance against overfitting, as shown in Clinical BioBERT's fast and clear feature disambiguation.

## 6   INTERPRETATION OF THE FINE-TUNED FEATURE SPACE

In this section, we perform outlier analysis on the fine-tuned feature spaces of the models yielding insights into their failure modes. Our first observation is that the feature spaces undergo extensive sparsification through fine-tuning. Subsequently, we leverage this structure to identify outlier reports in the feature space and solicit expert evaluation to determine the causes of this behavior. We demonstrate that different pre-trained models exhibit qualitative differences in their outlier modes and SUFO provides useful practical insight into the behavior of these models under fine-tuning.

### 6.1   THE STRUCTURE OF THE FINE-TUNED FEATURE SPACE

We analyze principle components (PCs) of features in the final layer classification token of the fine-tuned models. These features are important as they are used directly for prediction, and often contribute the most to performance in ablation studies (Merchant et al., 2020; van Aken et al., 2019).

**High sparsity**   We first show that the fine-tuned last layer classification token feature space is highly sparsified. We observe that, for every model across the four pathology tasks, the first two PCs explain on average 95% of the variance in the dataset. To understand how the PCs contribute to model performance, we conduct a PC probing experiment (see Appendix A.7). In the experiment, we measure model performance on reconstructed rank-$k$ feature space by projecting onto the bottom $k$ PCs, with $k$ varying between 1 and 768. In particular, $k = 768$ corresponds to the full-feature space. In the PC probing result, we see the first 2 PCs contribute significantly to model performance from the surge in the performance after adding them back in at $k = 767$ and $k = 768$.

**Outlier extraction**   The sparsified low-dimensional structure now allows us to inspect and interpret the fine-tuned features. As we will see,  we do this by interpreting outlier reports that do not conform to the *typical* behavior of an input report. To extract these outliers, we construct clusters of the training set[11] on the two-dimensional singular subspace of the feature space. We observe a strong clustering phenomenon where most samples cluster based on their labels (see Figure 2). The main difficulties in extracting these clusters are that the one-dimensional projections onto PC1 and PC2 often exhibit significant differences in scale and distribution across the four tasks (Figure 2b) and furthermore, the cluster structure itself is also correspondingly different (Figure 2b) across tasks. To

---

[11]We choose the training set here to ensure there are a sufficient number of datapoints to reliably recover the PCs. This, however, limits our ability to examine the test set *generalization* of the models.

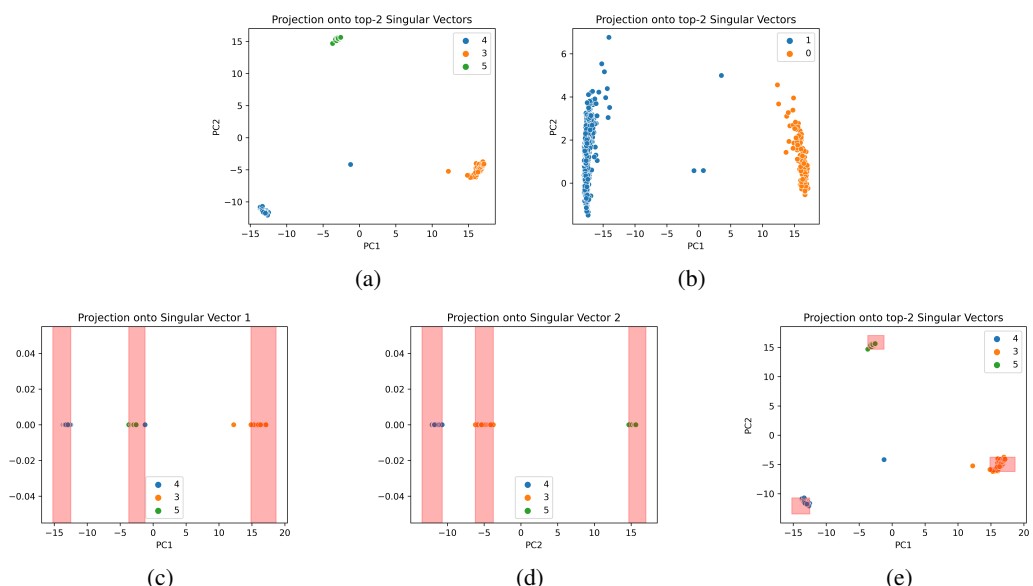

Figure 2: Illustration of clustering algorithm: fine-tuned BERT on Path-PG. (a) The projection onto the first two PCs. (b) A similar projection for Path-SV. Notice that the scales of the two PCs are drastically different and a naive clustering based on the Euclidean metric may not capture the variation in PC2. (c) The 3 clusters obtained solely from the projection onto PC1 with each red bar denoting the boundaries of a single cluster. (d) 3 clusters similarly obtained on PC2. (e) The final set of 3 clusters obtained by forming all possible combinations of clusters from (c) and (d) and selecting the three largest.

address this, we first independently extract clusters for the one-dimensional projections onto PC1 (either 2 or 3 depending on the number of labels) and PC2 (either 1 or 3). This produces intervals $\{I_{i,1}\}_{i=1}^{m_1}$ and $\{I_{i,2}\}_{i=1}^{m_2}$ where $m_1, m_2 \in [3]$ for PC1 and PC2 respectively. The clusters in the top-2 singular subspace are obtained by taking the cross product of all pairs of 1-dimensional clusters to obtain $m_1 \times m_2$ rectangles $\{I_{i,1} \times I_{j,2}\}_{i \in [m_1], j \in [m_2]}$. From these, the 2 or 3 (again depending on the number of labels) rectangles with the most datapoints are selected to obtain the final set of 2-dimensional clusters (represented by the red rectangles in Figure 2e). This process is illustrated in Figure 2. Finally, we extract as outliers all reports which do not fall into any of the clusters.

## 6.2 DOMAIN EXPERT EVALUATION

After extracting these outliers, we solicited feedback from a domain expert (a clinician in Urology) to attempt to explain their behavior. They were asked whether an outlier report would be challenging for human classification, and if so, explain why. The models were then compared on this feedback.

**Common outlier modes** We identified the following common outlier modes from our expert-provided feedback. Here, we restrict to the reports identified as being difficult to classify by our expert, henceforth referred to as Hard Outliers: (1) Wrongly labeled reports, (2) Inconsistent reports, (3) Multiple Sources of Information, (4) Not reported or truncated report, and (5) Boundary reports. A full description is provided in Table A5. The distribution of these classes of outlier reports for each model is provided in Table A6. The main difference between models is their sensitivity to truncated/unreported (4) instances. In general, Clinical BioBERT and PubMedBERT identify more instances where the target label is not present than BERT, BioBERT, and TNLR. Hence, the sparsified feature spaces of PubMedBERT and Clinical BioBERT allow for improved detection of missing *medical information* in the pathology reports. We believe the two models extract more comprehensive features that better model the medical data than their general counterparts. On the other hand, features extracted by PubMedBERT are less robust leading to overfitting during fine-tuning. We at-

tribute the inferior performance of the other mixed-domain model, BioBERT, compared to Clinical BioBERT to the lack of clinical data in its pre-training corpus.

# 7 CONCLUSIONS

In this work, we developed SUFO, a systematic pipeline to shed light on the fine-tuned feature spaces of transformers for increased interpretability by domain practitioners, helping ensure trust in and safety of LMs in critical application domains such as medicine. In our case study investigating the impact of pre-training data on fine-tuned features for clinical note classification, we reveal the robustness of mixed-domain models under substantial class imbalance, that in-domain pre-training helps faster feature disambiguation, and improved identification of missing medical information, validated by an expert evaluation. Although this work takes positive steps towards transparent LMs for medicine, our results are limited in scale and to the setting of the clinical classification tasks. More work is needed to generalize these findings to a broader set of clinical tasks and models. Finally, optimally combining a large general-domain corpus and a smaller domain-specific one for effective pre-training is an important direction for future work.

ACKNOWLEDGMENTS

This work is partially supported by NSF DMS-grant 2015341 and NSF grant 2023505 on Collaborative Research: Foundations of Data Science Institute (FODSI). ARH was supported by the Chancellor's Fellowship from University of California, Berkeley. We thank Peter Potash for insightful discussions at the development stage of this work, and the Microsoft Turing Academic Program team for access to the TNLR model.

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

# A  APPENDIX

## A.1  DESCRIPTION OF EXTRACTED PATHOLOGIC DATA ELEMENTS

Table A1: Description of the 4 extracted pathologic data elements.

| Data elements | Description |
| --- | --- |
| Primary Gleason grade | A whole number from 1 to 5 representing the primary score given to a specimen based on the Gleason grading system to measure tumor aggressiveness. |
| Secondary Gleason grade | A whole number from 1 to 5 representing the secondary score given to a specimen based on the Gleason grading system to measure tumor aggressiveness. |
| Margin status for tumor | To evaluate surgical margins, the entire prostate surface is inked after removal. The surgical margins are designated as "negative" if the tumor is not present at the inked margin, and "positive" if tumor is present. |
| Seminal vesicle invasion | Invasion of tumor into the seminal vesicle. It is marked as "negative" if no invasion is present in the seminal vesicle, and "positive" if invasion is present. |

## A.2  ANONYMIZED PATHOLOGY REPORT SAMPLES

- synoptic comment for prostate tumors " 1.  type of tumor
  :  adenocarcinoma small acinar type.  " 2.  location of
  tumor :  both lobes.  3.  estimated volume of tumor :
  3.  5 ml.  4.  gleason score :  4 + 3 = 7.  5.  estimated
  volume > gleason pattern 3 :  2 ml.  6.  involvement of
  capsule :  present ( e.  g.  slide b6 ).  7.  extraprostatic
  extension :  not identified.  8.  status of excision margins
  for tumor :  negative.  status of excision margins for
  benign prostate glands :  positive ( e.  g.  slide b4 ).
  9.  involvement of seminal vesicle :  not identified.  10.
  perineural infiltration :  present ( e.  g.  slide b11 ).  "
  11.  prostatic intraepithelial neoplasia ( pin ) :  present
  high – grade ( e.  g " slide b4 ).  12.  ajcc / uicc stage
  :  pt2cnxmx ; stage ii if no metastases are identified.
  13.  additional comments :  none.  final diagnosis :  " a.
  prostate left apical margin :  benign prostatic tissue.  "
  " b.  prostate and seminal vesicles resection :  prostatic
  adenocarcinoma " gleason score 4 + 3 = 7 ; see comment.
- synoptic comment for prostate tumors – type of tumor :
  small acinar adenocarcinoma.  – location of tumor :  – right
  anterior midgland :  slides b3 – b5.  – right posterior
  midgland :  slides b6 – b8.  – left anterior midgland :
  slides b12 – b14.  – left posterior midgland :  slides b9
  – b11.  – left and central bladder bases :  slides b16 – b17
  – estimated volume of tumor :  10 cm3.  " – gleason score :
  7 ; primary pattern 3 secondary pattern 4.  " – estimated
  volume > gleason pattern 3 :  40 %.  " – involvement of
  capsule :  tumor invades capsule but does not extend beyond
  " " capsule ( slides b5 b8 b18 ).  " – extraprostatic
  extension :  none.  – margin status for tumor :  negative.
  – margin status for benign prostate glands :  negative.  –
  high – grade prostatic intraepithelial neoplasia ( hgpin

```
) : present ; extensive. - tumor involvement of seminal
vesicle : none. - perineural infiltration : present. -
lymph node status : none submitted. - ajcc / uicc stage :
pt2cnx. final diagnosis : " a. prostate left base biopsy
: fibromuscular tissue no tumor. " " b. prostate radical
prostatectomy : " " 1. prostatic adenocarcinoma gleason
grade 3 + 4 score = 7 involving " " bilateral prostate
negative margins ; see comment. 2. " seminal vesicles with
no significant pathologic abnormality.
```

## A.3 FINE-TUNING

We fine-tune the models to perform single-label classification for all tasks. We add a linear layer followed by a softmax function to the model output on the classification token. The datasets are divided into 71% training, 18% validation, and 11% test, with label distribution in each set resembling the distribution in the full datasets. Best model checkpoints are selected based on validation set performances, and are used in all experiments. For pathology reports, we evaluate the models against macro F1 as each class accounts for equal importance, while we report accuracy for MedNLI. We set the encoder sequence length to 512 tokens for pathology reports, and 256 tokens for MedNLI, which allows us to encode the full length of the majority of the datasets.

**Prostate Cancer Pathology Reports** We use consistent fine-tuning hyperparameters for all models and all the four tasks, as we observe the validation set performance is not very sensitive to hyperparameter selection (less than 1% F1 performance change). We use an AdamW optimizer with a $7.6 \times 10^{-6}$ learning rate, 0.01 weight decay, and a $1 \times 10^{-8}$ epsilon. We also adopt a linear learning rate schedule with a 0.2 warm-up ratio. We fine-tune for a maximum of 25 epochs with a batch size of 8 and evaluate every 50 steps on the validation set. Each model is fine-tuned on a single NVIDIA Tesla K80 GPU, and average fine-tuning time is around 3 hours.

**MedNLI** We use consistent fine-tuning hyperparameters for all models, as we observe the validation set performance is not very sensitive to hyperparameter selection (less than 1% accuracy change). We use an AdamW optimizer with a per-layer learning rate decay schedule ($1 \times 10^{-4}$ as the starting learning rate, and 0.8 as the decay factor), 0 weight decay, $1 \times 10^{-6}$ epsilon, and a 0.1 warm-up ratio. We fine-tune for a maximum of 10 epochs with a batch size of 32 and evaluate every epoch on the validation set. Each model is fine-tuned on a single NVIDIA GeForce GTX TITAN X GPU, and the fine-tuning time on average is less than 1 hours.

## A.4 PER-CLASS ACCURACY ON PATH-PG AND PATH-SG

Table A2: Per-class accuracy of the five models on Path-PG and Path-SG, averaged across three runs (all stds are $< 5\%$ so we omit it to save spaces). PubMedBERT performs poorly when classifying the minority class 5 in the highly imbalanced Path-PG dataset, while it obtains descent performance across all classes in the slightly more balanced Path-SG dataset.

| | Path-PG | | | Path-SG | | |
|---|---|---|---|---|---|---|
| Models \Labels | 3 | 4 | 5 | 3 | 4 | 5 |
| BERT | 0.99 | 0.94 | 1.00 | 0.98 | 0.98 | 0.97 |
| TNLR | 0.97 | 0.87 | 1.00 | 0.99 | 0.99 | 0.99 |
| BioBERT | 0.99 | 0.97 | 0.94 | 0.99 | 0.99 | 0.99 |
| Clinical BioBERT | 0.99 | 0.98 | 1.00 | 0.99 | 0.99 | 0.98 |
| PubMedBERT | 0.99 | 0.92 | **0.67** | 0.98 | 0.99 | 0.97 |

## A.5 FINE-TUNING RESULTS ON MEDNLI

Table A3: Per-class accuracy and overall accuracy of PubMedBERT and Clinical BioBERT on MedNLI across three runs, where three scenarios are evaluated: Balanced ('C':'E':'N'=34%:33%:33%), Imbalanced ('C':'E':'N'=39%:53%:8%), and Highly Imbalanced ('C':'E':'N'=67%:30%:3%).

| Labels \Models | Balanced | | Imbalanced | | Highly Imbalanced | |
|---|---|---|---|---|---|---|
| | PubMedBERT | Clinical BioBERT | PubMedBERT | Clinical BioBERT | PubMedBERT | Clinical BioBERT |
| Contradiction ('C') | 0.88 (0.03) | 0.76 (0.03) | 0.76 (0.03) | 0.70 (0.02) | 0.80 (0.01) | 0.79 (0.03) |
| Entailment ('E') | 0.75 (0.02) | 0.71 (0.02) | 0.71 (0.03) | 0.70 (0.02) | 0.34 (0.02) | 0.62 (0.05) |
| Neutral ('N') | 0.77 (0.05) | 0.72 (0.01) | 0.33 (0.16) | 0.32 (0.01) | 0.04 (0.03) | 0.04(0.02) |
| Accuracy | 0.83 (0.01) | 0.73 (0.01) | 0.70 (0.02) | 0.71 (0.01) | 0.71 (0.01) | 0.76 (0.03) |

## A.6 QUANTIFYING THE CLOSENESS BETWEEN PRE-TRAINING DATA AND TARGET DATA

We use perplexity of pre-trained models on target tasks to define the closeness between pre-training data and target data. The lower the perplexity means the closer the two data distributions should be.

Table A4: Perplexity of the five models on pathology reports.

| | BERT | TNLR | BioBERT | Clinical BioBERT | PubMedBERT |
|---|---|---|---|---|---|
| Perplexity | 1.111 | 1.115 | 1.113 | 1.110 | 1.103 |

## A.7 AUXILIARY MATERIAL FOR SECTION 6.1

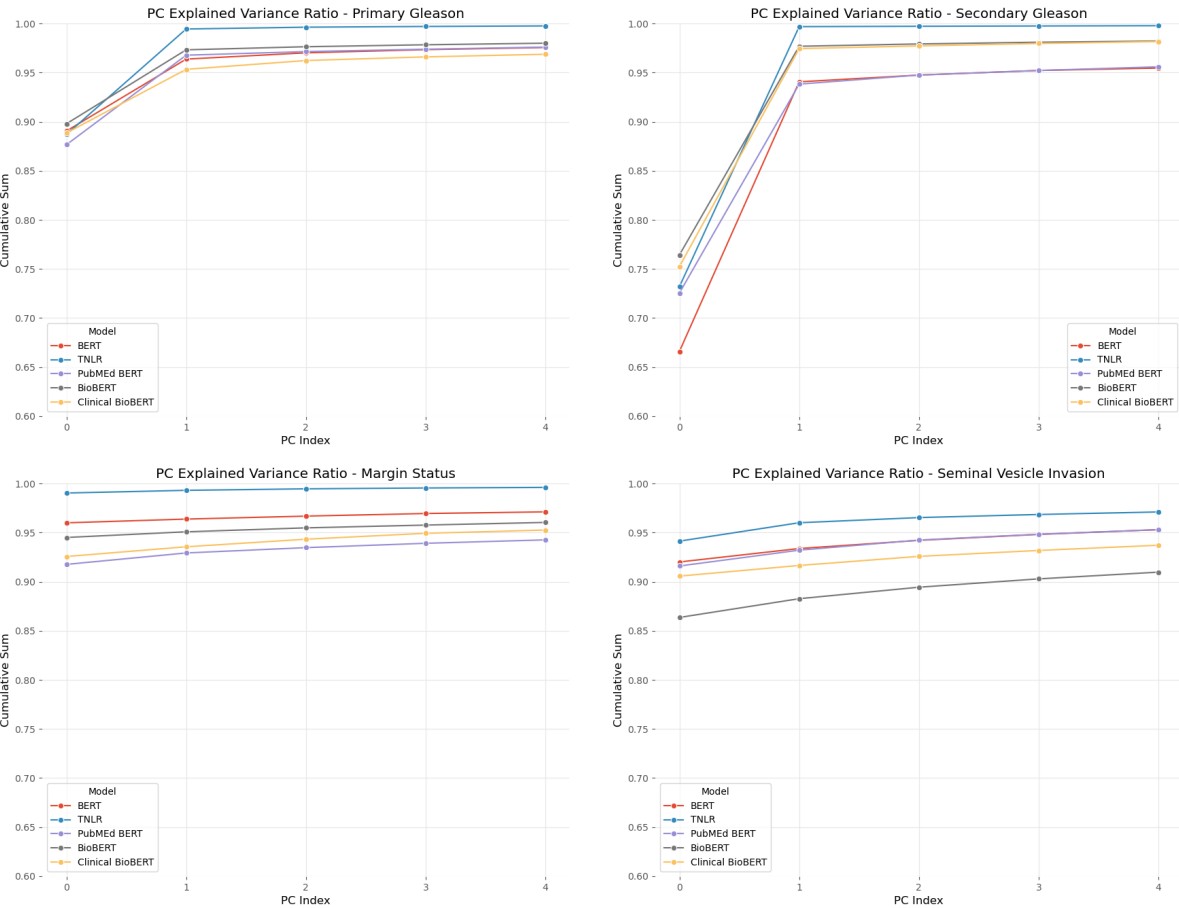

Figure A1: The first two PCs in the fine-tuned last layer classification token feature spaces of all the models explain on average 95% of the dataset variance across the 4 tasks.

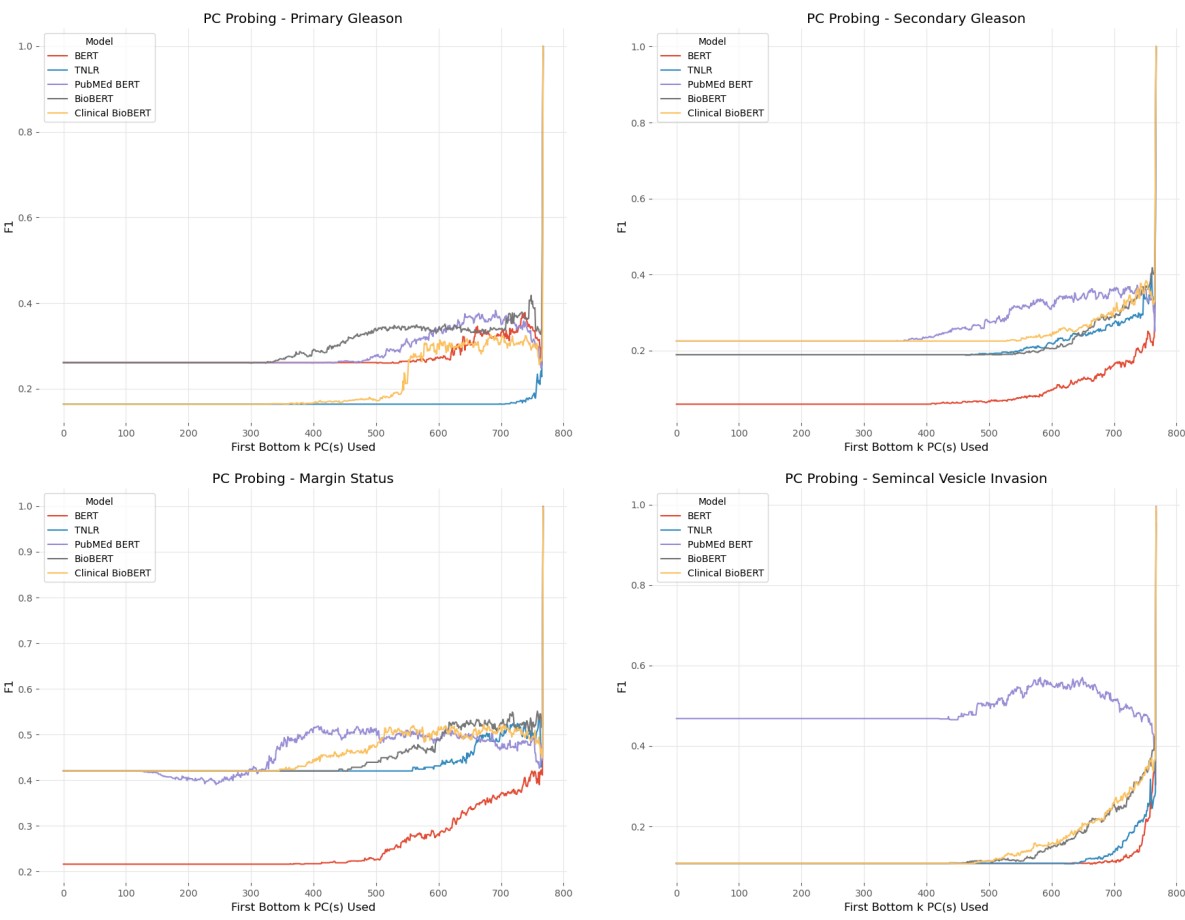

Figure A2: Filling back in the first two PCs, at the last two steps, $k = 767$ and $k = 768$, yields significant model performance gain.

## A.8    AUXILIARY MATERIAL FOR SECTION 6.2

The full categorization of the types of outliers obtained from our expert evaluation is provided in Table A5, while the distribution of these classes of outlier reports for each model are provided in Table A6.

Table A5: Description of categories of hard outliers

| Outlier Category ID | Category Name | Category Description |
|---|---|---|
| 1 | Wrongly labeled report | These are reports for which the provided annotation is incorrect. For example, a report with `null` Gleason score corresponding to a scenario where a Gleason score cannot be assigned is wrongly included with another label. |
| 2 | Inconsistent report | For these reports, there exists inconsistent declarations of the target attribute (say, Primary Gleason score) in two different parts of the report. |
| 3 | Multiple Sources of Information | These reports contain multiple sources of information which are composed to produce one final label. One such instance of such an outlier (for the Secondary Gleason label) contained scores from five tumor nodules which were then combined to give one final composite score. A classifier must learn to distinguish the true final score from those that were used to obtain it. |
| 4 | Not reported or truncated report | These are reports for which the target attribute is either not reported or the report is truncated before entry into the database. |
| 5 | Boundary reports | These reports feature scenarios where the target attribute is hard to determine precisely or requires some interpretation of the provided information. For instance, one such report presents a Gleason score with a combined value of 7 with the other information in the report requiring the classifier to deduce that the Gleason score is $3 + 4$. |

Table A6: A distribution of Hard Outliers for each model categorized according to the 5 outlier types.

| Outlier Type | BERT | BioBERT | Clinical BioBERT | PubMedBERT | TNLR |
|---|---|---|---|---|---|
| 1 | 0 | 0 | 1 | 1 | 1 |
| 2 | 0 | 1 | 0 | 1 | 2 |
| 3 | 2 | 0 | 1 | 1 | 1 |
| 4 | 0 | 1 | 3 | 5 | 1 |
| 5 | 4 | 0 | 3 | 3 | 2 |
| Total | 6 | 2 | 8 | 11 | 7 |

## A.9 FEATURE DYNAMICS

Here we present comprehensive sets of feature scatterplots along layers 1 to layer 12 (top-down) and selected epochs in the order of $1, 2, 3, 4, 5, 6, 7, 8, 9, 10, 15, 20, 25$ (left-right) of the 5 models, as we observe the models typically show the most rapid performance gain from epoch 1 to 10, and marginal increase afterwards. We include the plots from Path-PG and Path-MS, as representatives of tasks having different number of labels to save space, but note that we observe similar trend in the results of all the 4 tasks

### A.9.1 PATH-PG

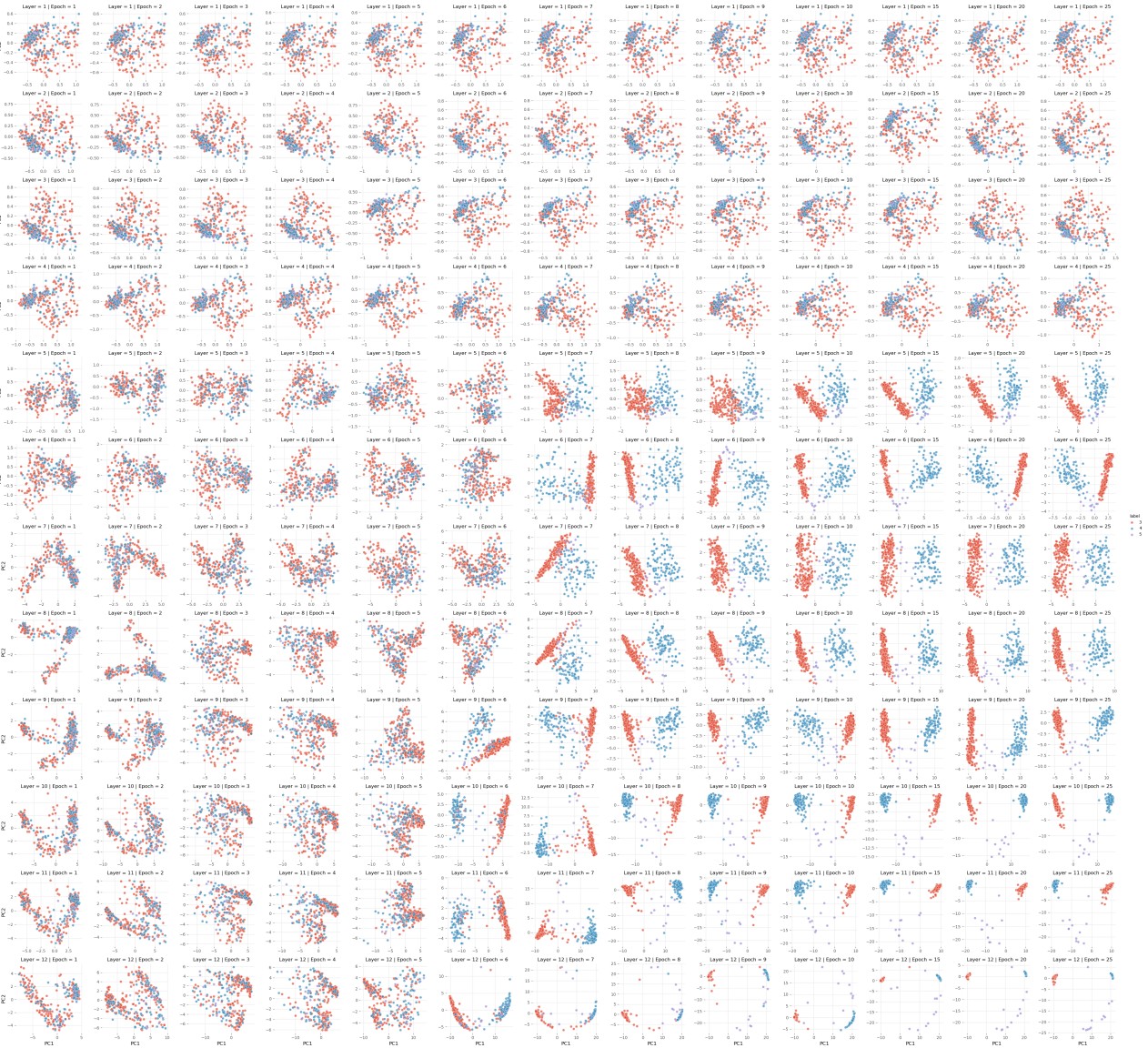

Figure A3: Path-PG: BERT

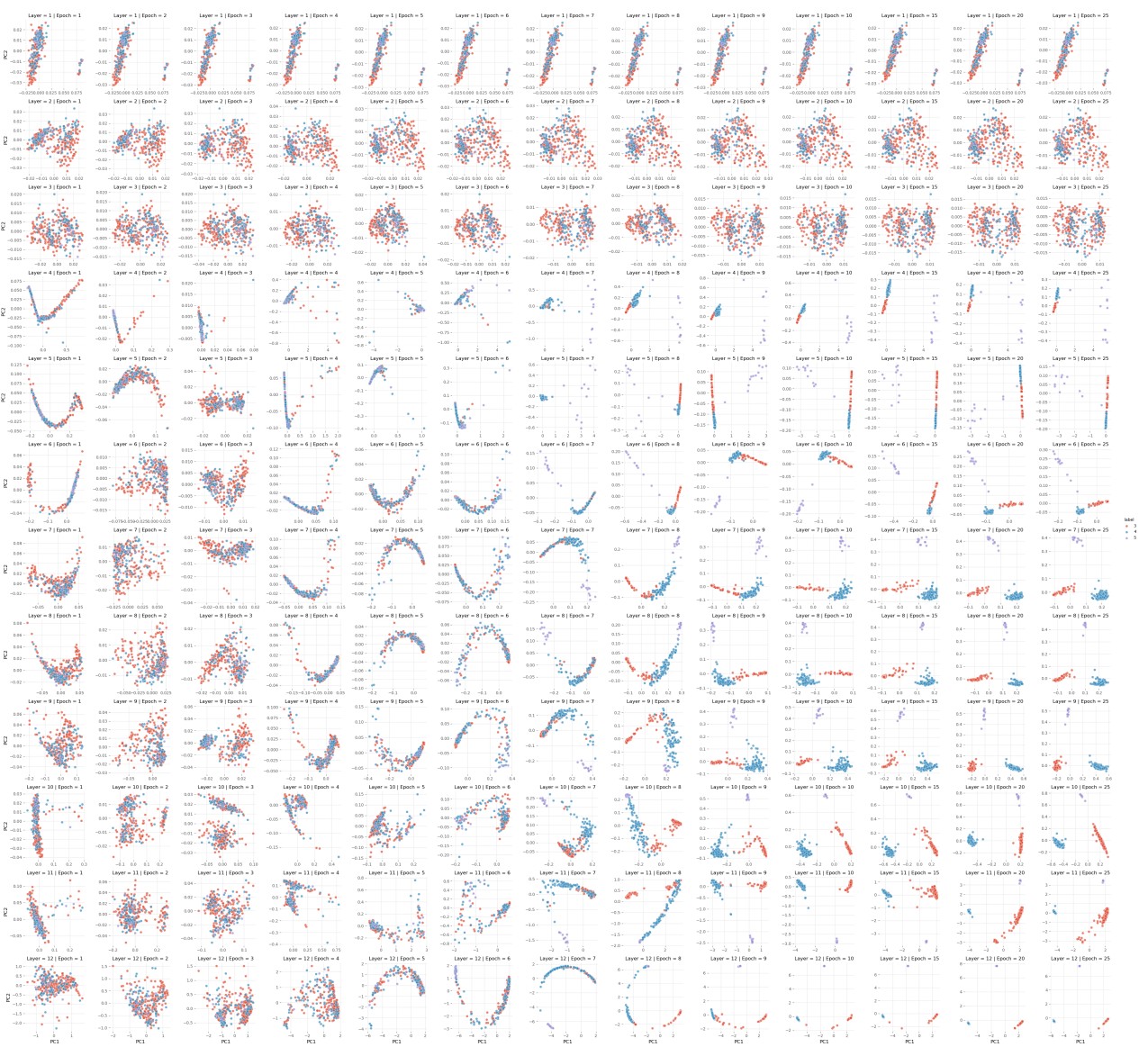

Figure A4: Path-PG: TNLR

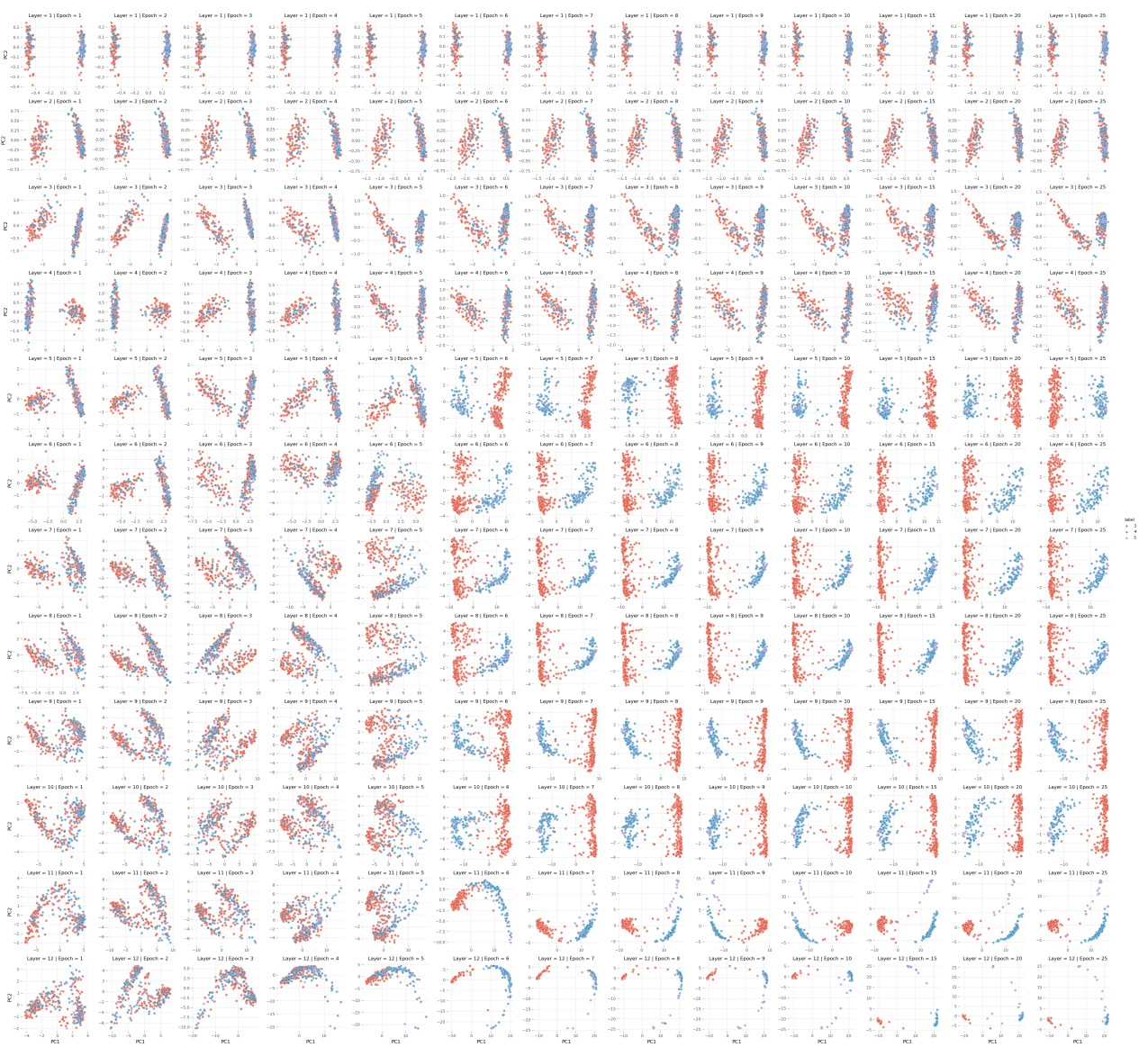

Figure A5: Path-PG: BioBERT

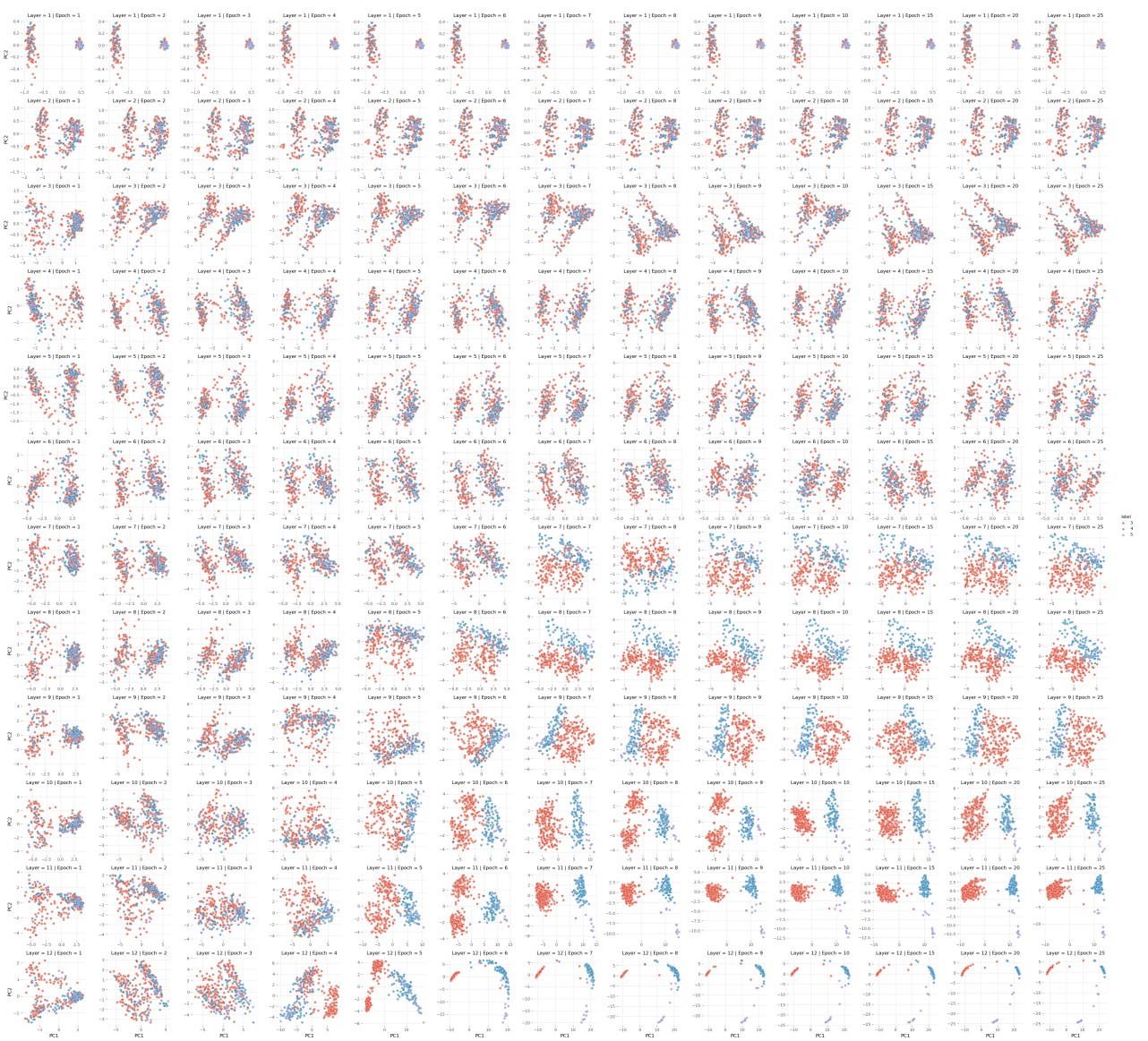

Figure A6: Path-PG: Clinical BioBERT

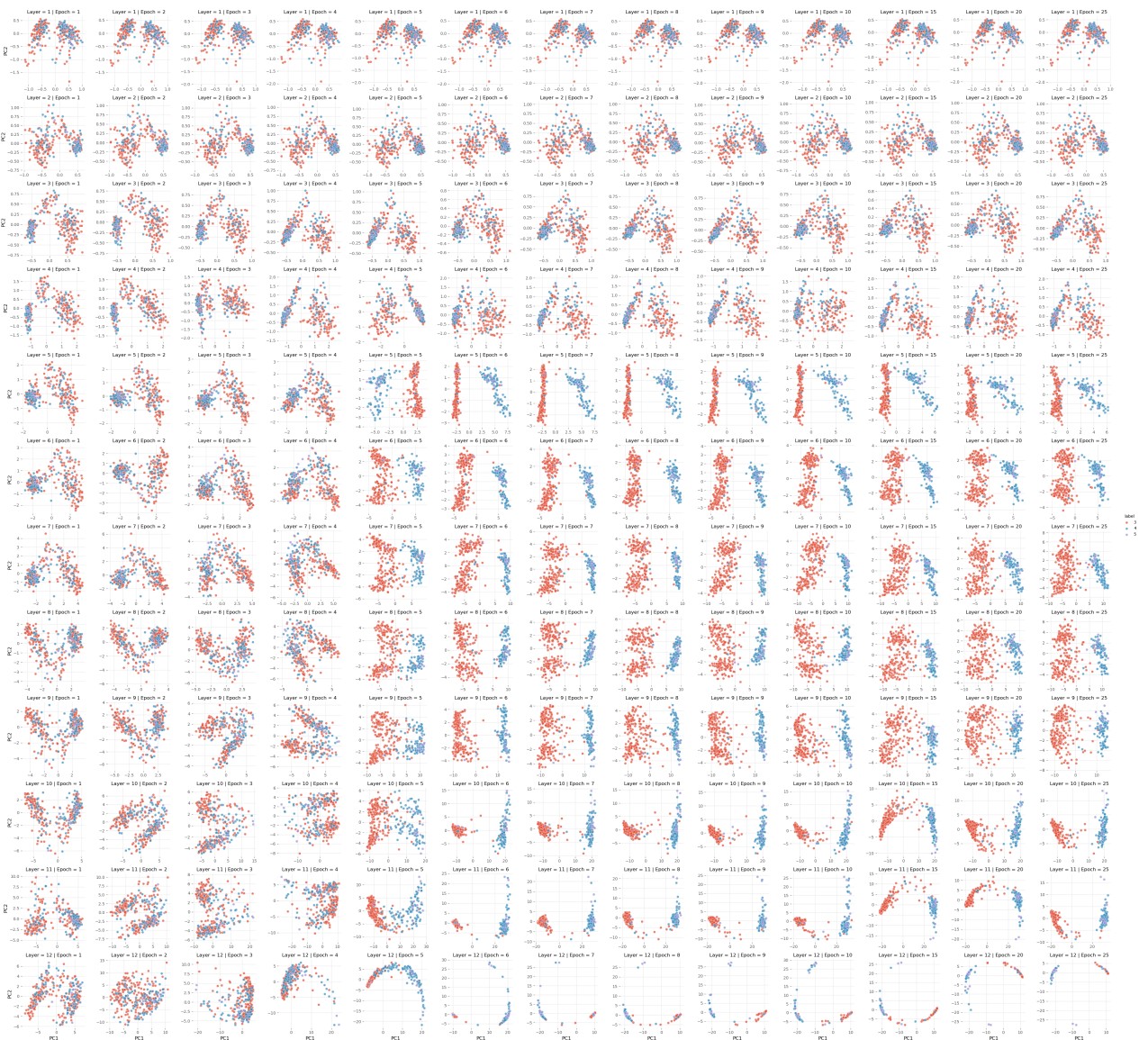

Figure A7: Path-PG: PubMedBERT

### A.9.2 PATH-MS

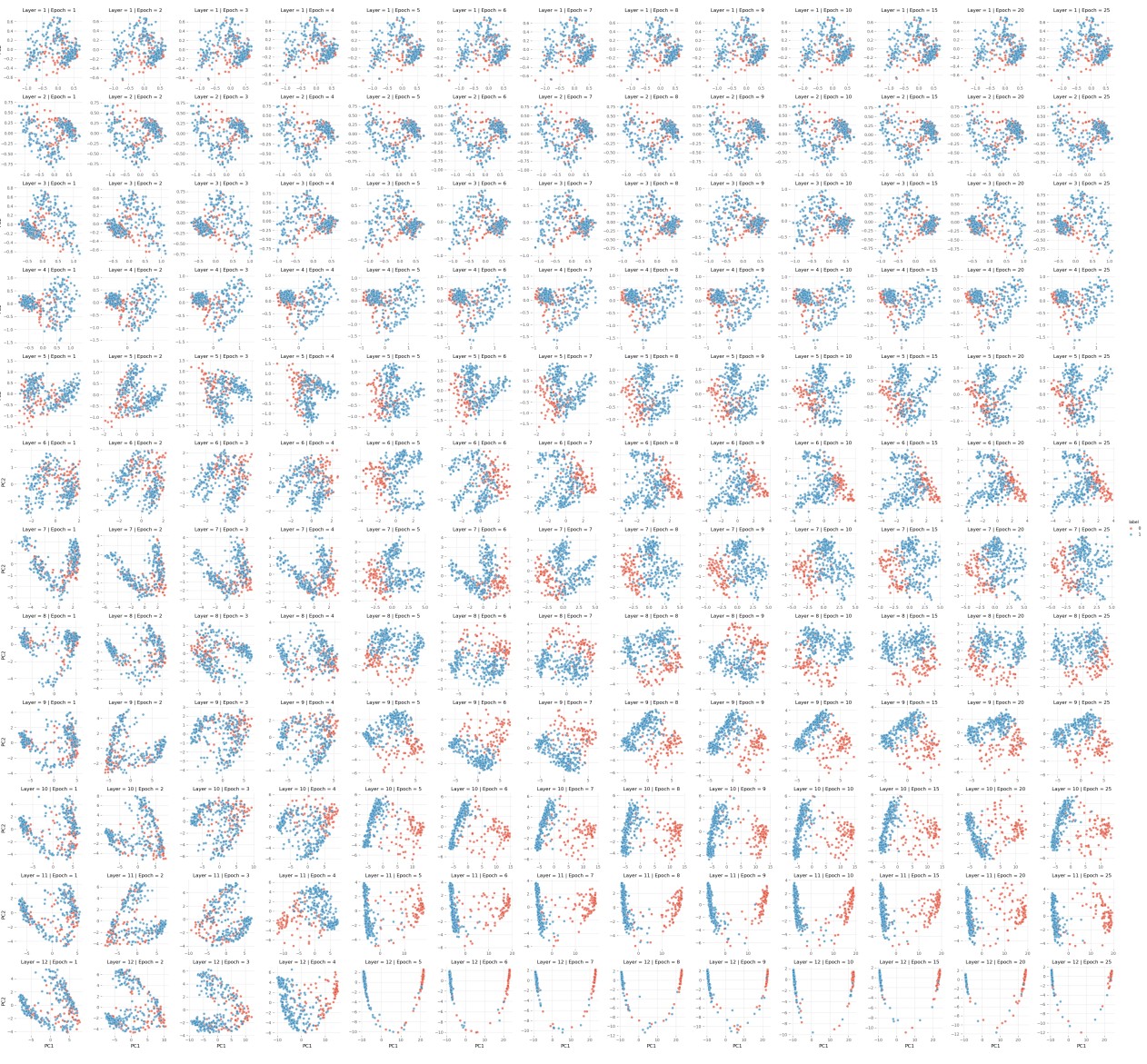

Figure A8: Path-MS: BERT

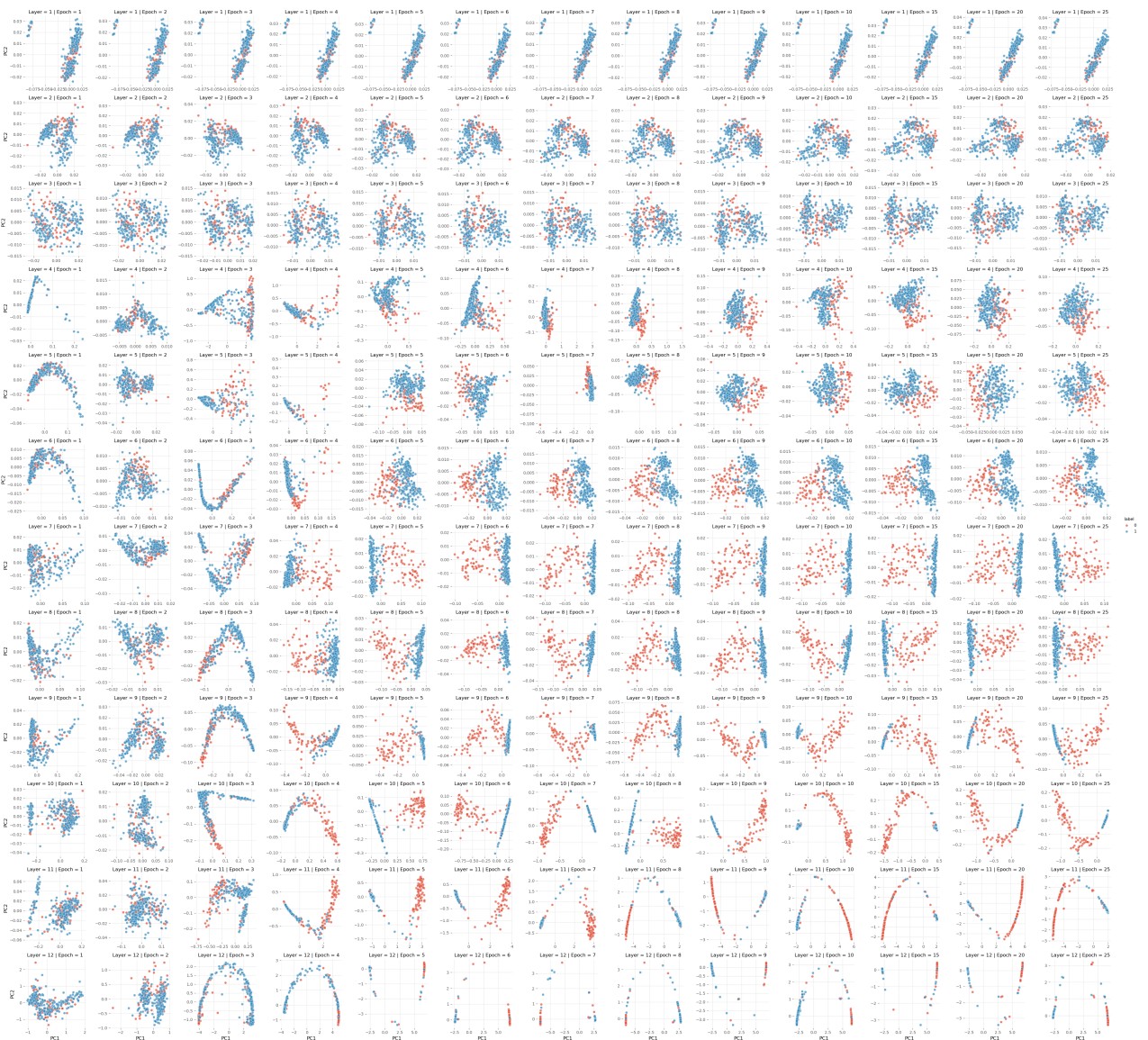

Figure A9: Path-MS: TNLR

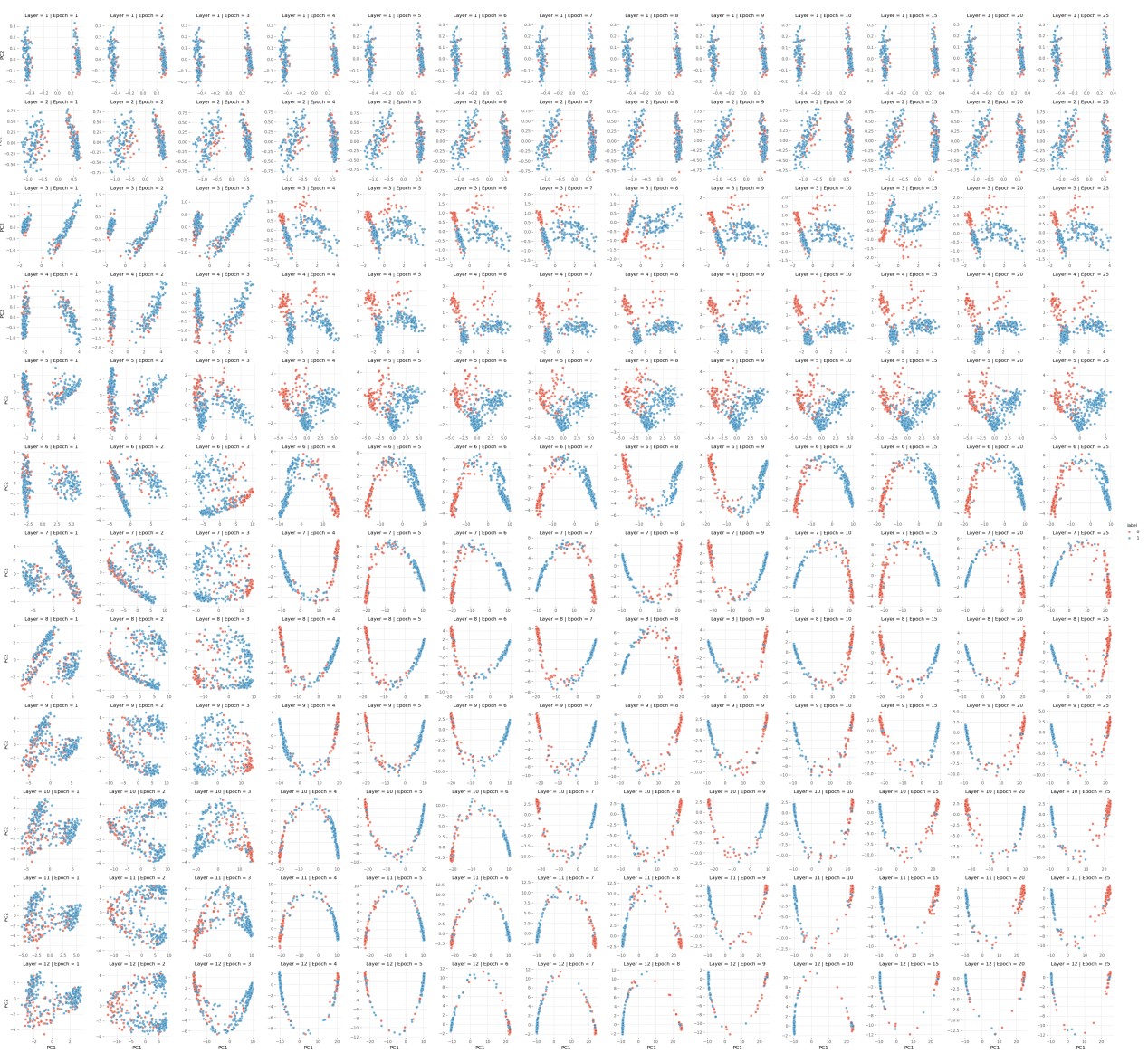

Figure A10: Path-MS: BioBERT

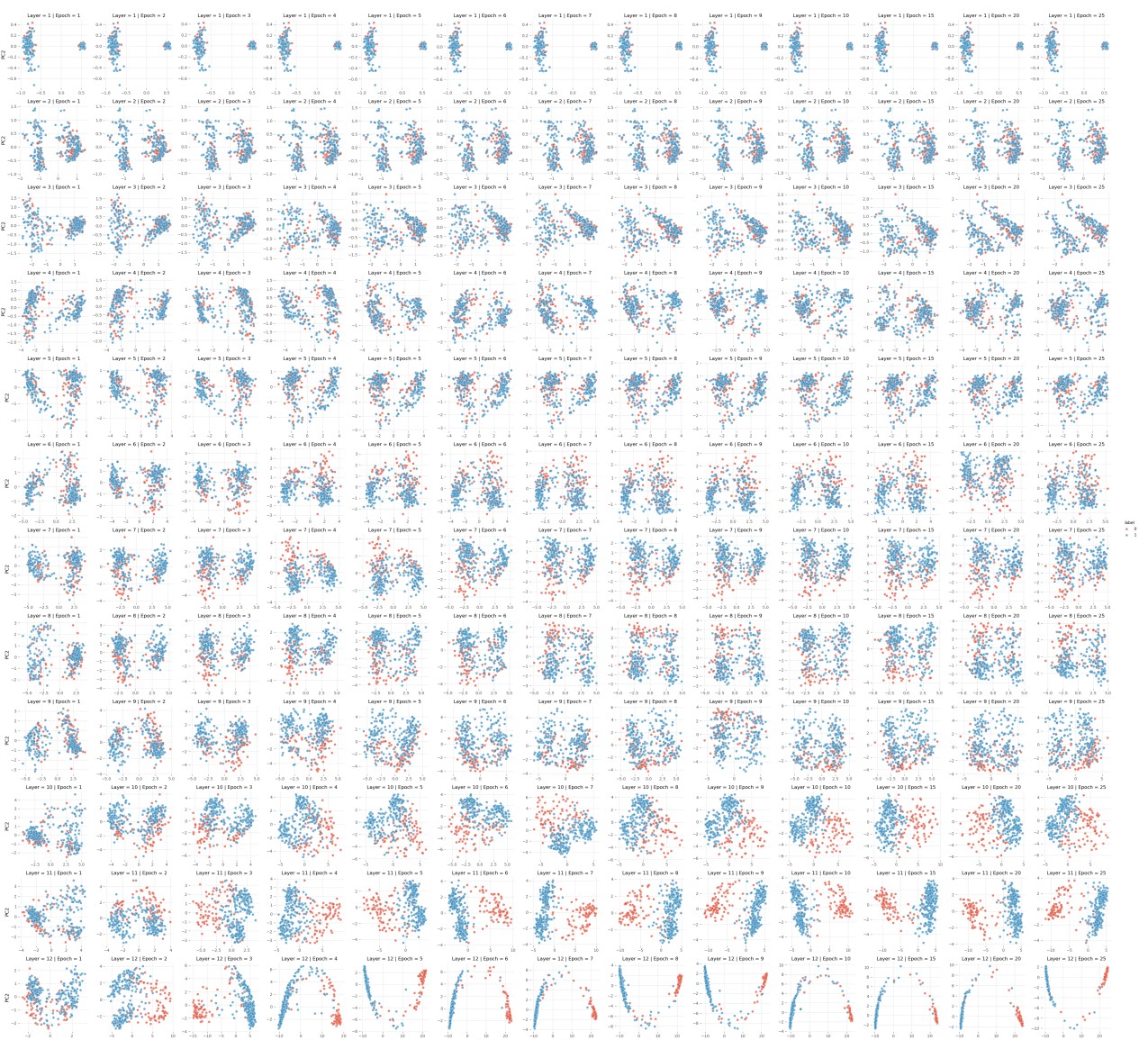

Figure A11: Path-MS: Clinical BioBERT

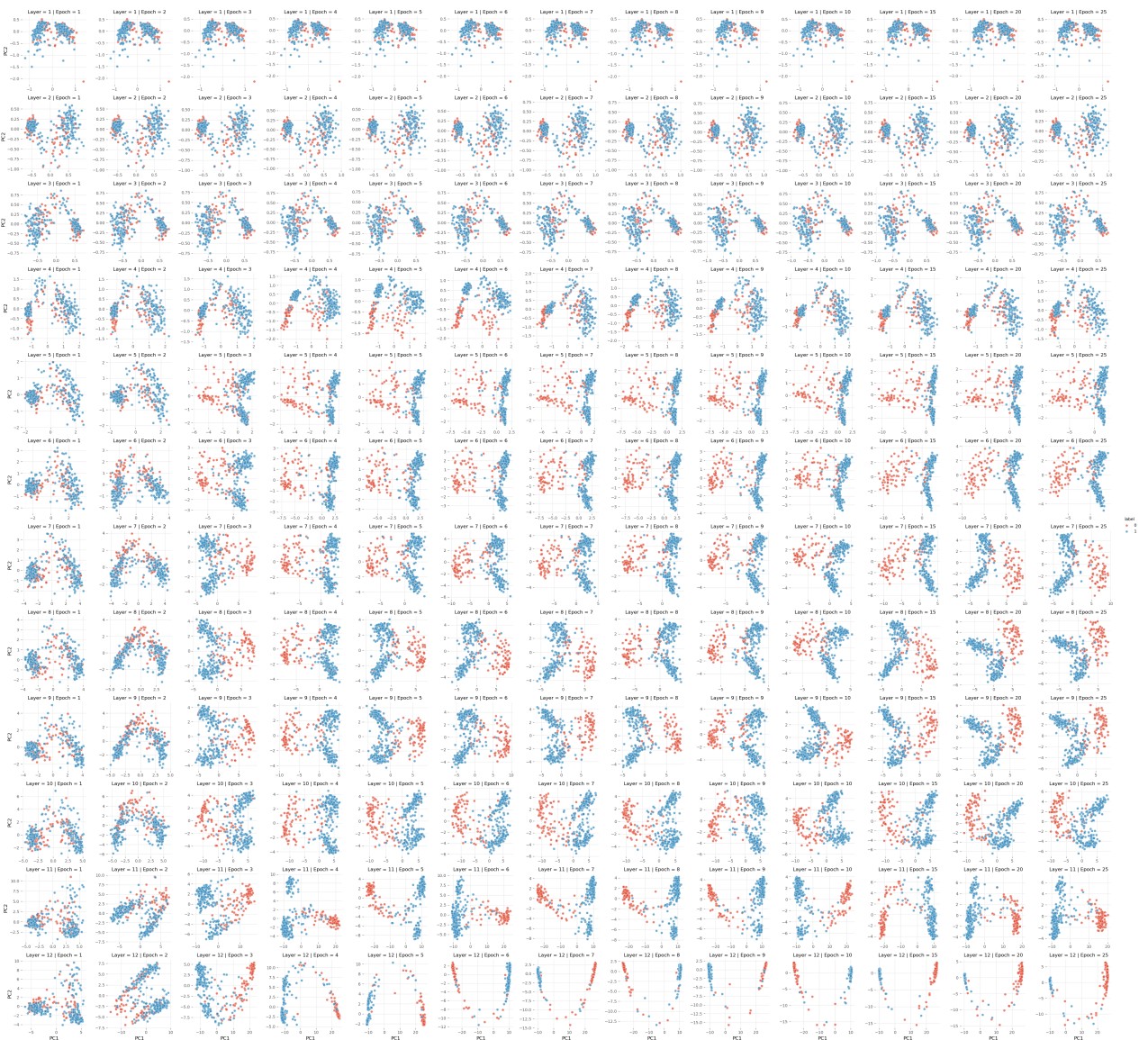

Figure A12: Path-MS: PubMedBERT

