# OpenReview forum: "Diagnosing Transformers: Illuminating Feature Spaces for Clinical Decision-Making"
_ICLR.cc/2024/Conference — ICLR 2024 poster_

### Official Review · Reviewer_bnKW · 2023-10-28

**Soundness:** 3 good
**Presentation:** 3 good
**Contribution:** 3 good
**Rating:** 6
**Confidence:** 4

**Summary:**

This work provides a comprehensive framework for analyzing the pretrained and fine-tuned feature spaces of clinical transformers. The provided analyses include supervised probing, unsupervised similarity analysis, feature dynamics, and outlier analysis.

**Strengths:**

The authors evaluate five pre-trained transformer models on real-world pathology tasks, offering a multifaceted perspective on the subject. The paper is generally well-written, and the results provide insights for researchers in the medical NLP field.

**Weaknesses:**

My major comments are:

1. The choice to evaluate the results predominantly on a private dataset, rather than well-established clinical NLP datasets such as MIMIC-III, raises concerns about the reliability and generalizability of the drawn conclusions. Although the authors did incorporate the public MedNLI dataset, they pointed out certain imbalance issues. The principal analyses still predominantly focus on the private pathology dataset.

2. The paper contains an extensive number of tables and figures. It is noted that even some key conclusions are presented in the Appendix. It might be beneficial for the authors to relocate some of the less important content to the Appendix and highlighting major results within the main text.

3. The section evaluating clinical practitioners' perspectives appears to only incorporate feedback from a single practitioner, which is hard to support the claim that the findings in this work increase the interpretability to domain practioners.

**Questions:**

Please see the weaknesses above.

---

> ### Author Response · Authors · 2023-11-19
>
> Thanks for the helpful comments and thoughtful feedback (and large time commitment). We sincerely appreciate your critique and address your concerns below.
>
> | 1. The choice to evaluate the results predominantly on a private dataset, rather than well-established clinical NLP datasets such as MIMIC-III, raises concerns about the reliability and generalizability of the drawn conclusions. Although the authors did incorporate the public MedNLI dataset, they pointed out certain imbalance issues. The principal analyses still predominantly focus on the private pathology dataset.
>
> Thank you for the comment! However, please note that any datasets (e.g. MIMIC, PubMed) included in the pre-training data of either models we evaluated would not be suitable to be used in the experiments. Regardless, we realize the importance of generalizability, and additionally report our results on MedNLI, a balanced public dataset from a well-acknowledged biomedical benchmark, BLURB. MedNLI is by nature a balanced dataset. In the fine-tuning experiments, we deliberately manipulate the label ratios in MedNLI to simulate different scenarios we have in the real-world pathology report data, and are able to show comparable results on the two datasets. From this, we chose to carry out subsequent steps of SUFO on pathology report data as a case study to showcase SUFO’s utility, but please note that SUFO is a task-agnostic and model-agnostic pipeline and can be easily applied to any data.
>
> | 2. The paper contains an extensive number of tables and figures. It is noted that even some key conclusions are presented in the Appendix. It might be beneficial for the authors to relocate some of the less important content to the Appendix and highlighting major results within the main text.
>
> Thanks for the suggestion. We will include condensed versions of these results in the main paper in a subsequent draft.
>
> | 3. The section evaluating clinical practitioners' perspectives appears to only incorporate feedback from a single practitioner, which is hard to support the claim that the findings in this work increase the interpretability to domain practioners.
>
> Indeed a good suggestion! We’ve reviewed the outlier analysis with another urologic oncologist, who agreed with the outliers identified based on the clinician’s feedback presented in the paper, and no changes were made from the oncologist’s feedback. In other words, the failure modes we identified for the pathology reports are believed to be general. We will include this in a subsequent draft.

---

> ### Comment · Reviewer_bnKW · 2023-11-20
>
> Thanks to the authors for their reply. I would stand for my score.

---

### Official Review · Reviewer_Y739 · 2023-11-01

**Soundness:** 4 excellent
**Presentation:** 4 excellent
**Contribution:** 4 excellent
**Rating:** 8
**Confidence:** 4

**Summary:**

The paper proposed a systematic framework, SUFO, to analyze and visualize the utility of five pre-trained transformers towards a pathology classification task, with validation on a public data MedNLI. SUFO is named after the following four aspects in diagnosing transformers: 1) Supervised probing 2) Unsupervised similarity analysis, 3) Feature dynamics visualization and 4) Outlier analysis.

**Strengths:**

- Originality: the paper proposed a novel evaluation/diagnosing framework, that combines multiple existing ideas and approaches, for the application of five popular pre-trained transformer on a real-world data problem.
- Quality: the paper is well-organized with meaningful and inspiring research questions connecting each part of the paper. Technical analysis is sound with multiple metrics and various comparison aspects, along with intuitive plots to support the claims and conclusions. The hypothesis testing of whether domain-specific transformer will outperform general-trained BERT (even only on a single case study) is very interesting and inspiring.
- Clarity: the paper is easy to understand with nice writing flows throughout the problem construction, related work, experiment design, results discussion and conclusions. Assumptions and experiment design details are well-documented.
- Significance: the paper demonstrates solid results and meaningful discussions about application of pre-trained transformers/models for other researchers to refer to, test new hypotheses and build new experiments upon.

**Weaknesses:**

1. No significant technical weakness, but some weakness or unfulfilled audience expectations in terms of results inclusion/exclusion in the main paper. It is somehow disappointing when the highly expected results are in the Appendix, e.g. feature dynamics comparison mentioned in page 7, but in Appendix A.9 (is it possible to quantify the feature dynamics and compare them in plots like Figure 1, e.g. limit to first and last layer); sparsity comparison of top 2 PCs on page 8 but results in Appendix A.7 (same suggestion to include e.g. 1-2 plots of last layer), etc.
2. The potential application utility weakness of SUFO might lie in at the end of the transformer diagnosis towards a specific dataset, it is still unclear what transformer should an end user e.g. data scientist, choose from the five? How will SUFO provide robust and actionable insights based on the supervised and unsupervised analysis?

**Questions:**

1. Since the paper is based on a single datasets, could SUFO evaluate more benchmark datasets and data tasks to perform meta learning and automatically suggest the best pre-trained model given a dataset?

---

> ### Author Response · Authors · 2023-11-19
>
> Thanks for the helpful comments and thoughtful feedback (and large time commitment). We sincerely appreciate your critique and address your concerns below.
>
> | 1. No significant technical weakness, but some weakness or unfulfilled audience expectations in terms of results inclusion/exclusion in the main paper. It is somehow disappointing when the highly expected results are in the Appendix, e.g. feature dynamics comparison mentioned in page 7, but in Appendix A.9 (is it possible to quantify the feature dynamics and compare them in plots like Figure 1, e.g. limit to first and last layer); sparsity comparison of top 2 PCs on page 8 but results in Appendix A.7 (same suggestion to include e.g. 1-2 plots of last layer), etc.
>
> Thanks for the suggestion. We will include condensed versions of these results in the main paper in a subsequent draft.
>
> | 2. The potential application utility weakness of SUFO might lie in at the end of the transformer diagnosis towards a specific dataset, it is still unclear what transformer should an end user e.g. data scientist, choose from the five? How will SUFO provide robust and actionable insights based on the supervised and unsupervised analysis?
>
> This is a great question! We note that while SUFO provides insights into the behavior of various models under fine-tuning, the utilization of this information depends strongly on the nuances of the target application. Each step in SUFO sheds light on different parts of the fine-tuning model. The use of pre-trained probing helps evaluate the degree of overlap between the pre-trained data distribution which may be helpful in deciding which of the pre-trained models is most aligned with a target application while the evolution of the feature dynamics enables an understanding of the speed of the fine-tuning process over the candidate pre-trained models and finally, analyzing and visualizing the (sparsity properties of the) fine-tuned feature space is shown to be related to the sensitivities of the various models to different outlier modes. The choice of which model to use given the circumstances of a target application could vary widely. For instance, a model which through its feature dynamics analysis may be the fastest to fine-tune which may be important if there are a large number of tasks for which a classifier needs to be built. On the other hand, a more descriptive feature space in a fine-tuned model might allow a practitioner to diagnose dataset inconsistencies which require more careful human inspection to correctly evaluate. Despite these nuances, we believe that SUFO can aid in such practical decision making in several application domains by better illustrating the trade-offs between several pre-training choices.
>
> | 3. Since the paper is based on a single datasets, could SUFO evaluate more benchmark datasets and data tasks to perform meta learning and automatically suggest the best pre-trained model given a dataset?
>
> This is an interesting suggestion! As alluded to in the previous response, the nuances of a target application makes the design of a unifying framework for automatically suggesting an appropriate pre-trained model challenging. However, further development on the basic principles of SUFO and a more comprehensive evaluation on a broader range of tasks may enable such a procedure.

---

### Official Review · Reviewer_zoU3 · 2023-11-01

**Soundness:** 3 good
**Presentation:** 4 excellent
**Contribution:** 3 good
**Rating:** 6
**Confidence:** 3

**Summary:**

This paper conducts a case study on the Transformer model on clinical notes, including model fine-tuning, supervised probing, unsupervised similarity analysis, feature dynamics,  and outlier analysis. Results show several interesting points including the usefulness of mixed-domain models, in-domain pre-training helps faster feature disambiguation, and improved identification of missing medical information.

**Strengths:**

- The study offers a deep dive into understanding the interpretability of pre-trained Transformers within the crucial field of medical data.
- Most of the presented claims can be suppored by experiment results, shedding light on model selection, evaluation, and in-depth analysis processes.
- The paper is well written and easy to follow.

**Weaknesses:**

- The datasets used for fine-tuning are somewhat limited in their volume. Exploration with larger datasets, such as the MIMIC-IV clinical notes, might add more depth.
- While the author contends that PubMedBERT is brimming with pertinent data for the tasks at hand even before fine-tuning, the model displays unpredictability when predicting lesser-represented classes post-fine-tuning. Could enlarging the fine-tuning dataset address this challenge?
- The conclusion for Table 2 seems vague. PubMedBERT and BERT perform very similarly and it is hard to conclude that PubMedBERT contains much useful information for the tasks. Also, why the mixed-domian models BioBERT and Clinical BioBERT perform even worse than BERT?

**Questions:**

See weakness.

---

> ### Author Response · Authors · 2023-11-19
>
> Thanks for the helpful comments and thoughtful feedback (and large time commitment). We sincerely appreciate your critique and address your concerns below.
>
> | 1. The datasets used for fine-tuning are somewhat limited in their volume. Exploration with larger datasets, such as the MIMIC-IV clinical notes, might add more depth. While the author contends that PubMedBERT is brimming with pertinent data for the tasks at hand even before fine-tuning, the model displays unpredictability when predicting lesser-represented classes post-fine-tuning. Could enlarging the fine-tuning dataset address this challenge?
>
> Thank you for your comment! However, this work aims to showcase the utility of SUFO on realistic clinical use cases. Clinical data in reality comes in small sizes, ranging from a few hundreds to thousands. The size of the prostate cancer data (around 3000) is a good representative of the amount. In addition, please note that any datasets (e.g. MIMIC, PubMed) included in the pre-training data of either models evaluated would not be suitable to be used in our experiments. Regardless, we realize the importance of generalizability, and report our results on MedNLI, a public dataset with a larger size (around 7000 after sampling). From the comparable results that we see on the two datasets in the fine-tuning experiments, an increase in the dataset size didn’t seem to bring much noticeable difference as long as the data imbalance persists.
>
> | 2. The conclusion for Table 2 seems vague. PubMedBERT and BERT perform very similarly and it is hard to conclude that PubMedBERT contains much useful information for the tasks. Also, why the mixed-domian models BioBERT and Clinical BioBERT perform even worse than BERT?
>
> This is a very good point. The pre-trained features of BERT and PubMedBERT indeed perform comparably on average, with BERT having a slightly higher standard deviation across runs. In general, we believe this especially indicates the versatility of BERT’s feature for its ability to match models pre-trained using in-domain data (the versatility of BERT’s feature has also been much discussed in prior works on different NLP tasks. [1][2]) We will clarify this in the revision.
>
> [1] To Tune or Not to Tune? Adapting Pretrained Representations to Diverse Tasks. Matthew Peters et al.
>
> [2] A Feature Extraction based Model for Hate Speech Identification. Salar Mohtaj et al.

---

> > ### Comment · Reviewer_zoU3 · 2023-11-23
> >
> > Thanks to the authors for their reply. Regarding my first question, I think MIMIC-IV notes are not used in the training process for any of the five models (Clinical BioBERT is pre-trained on MIMIC-III notes but not MIMIC-IV). My other questions are clarified. I would keep my original score.

---

### Meta-Review · Area_Chair_uFVP · 2023-12-13

**Metareview:**

This well written paper has been assessed by three knowledgeable reviewers who voted for its acceptance (one full accept and two marginal accepts). The reviewers liked the originality and potential impact of the showcased work in healthcare, yet had minor concerns about generality of the conclusion since the method was demonstrated on only one publicly available dataset. But in summary, this work meets the requirements for acceptance at ICLR 2024.

**Justification For Why Not Higher Score:**

This is a good paper with some residual limitations, very well suited for a poster session.

**Justification For Why Not Lower Score:**

This paper is above the ICLR acceptance threshold.

---

### Decision · Program_Chairs · 2024-01-16

Accept (poster)